# Molecular patterns and mechanisms of tumorigenesis in HPV-associated and HPV-independent sinonasal squamous cell carcinoma

Fernando T. Zamuner [1,11], Sreenivasulu Gunti[2,11], Gabriel J. Starrett[3,11], Farhoud Faraji [4], Tiffany Toni[1], Anirudh Saraswathula[1], Kenny Vu [2], Anuj Gupta[5], Yan Zhang[5], Daniel L. Faden[6], Michael E. Bryan[6], Theresa Guo[4], Nicholas R. Rowan [1,7], Murugappan Ramanathan Jr.[1,7], Andrew P. Lane [1], Carole Fakhry[1], Gary L. Gallia[1,7,8], Clint T. Allen [9], Lisa M. Rooper[10] & Nyall R. London Jr. [1,2,7,8] ✉

Mechanisms of tumorigenesis in sinonasal squamous cell carcinoma (SNSCC) remain poorly understood due to its rarity. A subset of SNSCC is associated with human papillomavirus (HPV), but it is unclear whether HPV drives tumorigenesis or acts as a neutral bystander. Here, we show that HPV-associated SNSCC shares mutational patterns found in HPV-associated cervical and head and neck squamous cell carcinoma, including lack of *TP53* mutations, hotspot mutations in *PI3K* and *FGFR3*, enrichment of APOBEC mutagenesis, viral integration at known hotspots, and frequent epigenetic regulator alterations. We identify HPV-associated SNSCC-specific recurrent mutations in *KMT2C*, *UBXN11*, *AP3S1*, *MT-ND4*, and *MT-ND5*, with *KMT2D* and *FGFR3* mutations correlating with reduced overall survival. We establish an HPV-associated SNSCC cell line, showing that combinatorial small-molecule inhibition of YAP/TAZ and PI3K synergistically suppresses clonogenicity. Combining YAP/TAZ blockade with vertical PI3K inhibition may benefit HPV-associated SNSCC, whereas targeting MYC and horizontal inhibition of RAS/PI3K may suit HPV-independent SNSCC.

Sinonasal squamous cell carcinoma (SNSCC) is an uncommon head and neck malignancy arising in the paranasal sinuses and nasal cavity. An incidence of ~0.4 cases per 100,000 persons per year and a male predilection of ~2:1 have been reported[1,2]. Smoking has been identified as a risk factor for SNSCC development, although to a lesser degree than for head and neck squamous cell carcinoma (HNSCC)[3]. Survival is poor with a 5-year survival rate of ~50%, which may be in part due to frequent advanced stage at presentation, invasion of adjacent neurovascular structures, and other contributing factors[1]. An improved understanding of SNSCC tumor biology is necessary to identify new potential molecular targets and to improve current therapeutic approaches.

Oropharyngeal squamous cell carcinoma (OPSCC) is a subset of HNSCC arising from the oropharynx and is well-known for its association with HPV. Similarly, SNSCC arises from the sinonasal tract, and understanding the parallels and distinctions among HNSCC subtypes including OPSCC and SNSCC is essential for elucidating the role of HPV in SNSCC tumorigenesis. ~25% of SNSCC are associated with the human

papillomavirus (HPV), compared to ~80% of OPSCC[4,5]. While HPV16 underlies ~90% of HPV-associated OPSCC, HPV-associated SNSCC is associated with HPV16 in only ~70% of cases, and other high-risk HPV subtypes are more frequently detected in SNSCC[5–9]. HPV-associated SNSCC appears to behave differently than HPV-associated OPSCC in key aspects. Firstly, while high rates of cervical node metastasis represent a hallmark of HPV-associated OPSCC, HPV-associated SNSCC displays very low rates of cervical metastasis[1]. Additionally, while HPV-positive tumor status confers a significant survival advantage to patients with OPSCC, this survival advantage is lower and less clear in HPV-associated SNSCC[10]. Finally, given the rarity of SNSCC compared to OPSCC, it remains unclear whether HPV is a driver in HPV-associated SNSCC tumorigenesis or merely a bystander infection or contaminant in the sinonasal cavity[1,11].

A variety of chromosomal aberrations have been noted in HPV-independent SNSCC including copy number variations, amplification, and microsatellite instability[12]. Targeted sequencing has found TP53 mutations in ~70% of SNSCC samples[13]. Our understanding of SNSCC tumor biology, however, remains surprisingly poor as genomics-wide analyses and high-throughput sequencing performed to date in SNSCC have been lacking. Indeed, to our knowledge only two cases of whole-exome sequencing of SNSCC have been reported as part of a larger HNSCC study[14,15]. Mutations in SYNE1 and NOTCH3 were noted in the HPV-independent SNSCC sample and, surprisingly, TP53 was the only mutation reported in the HPV-associated SNSCC sample[14,15]. The rarity of SNSCC represents an obstacle to accruing a large enough cohort for comprehensive genome-wide characterization thereby limiting our knowledge of mutational frequencies and mechanistic drivers of HPV-associated and HPV-independent SNSCC and underscores a critical barrier in the advancement of novel precision therapeutics for SNSCC.

We hypothesized that performing a comprehensive genome-wide characterization of clinically annotated cohorts of HPV-associated and HPV-independent SNSCC would reveal molecular patterns of tumorigenesis distinguishing these subtypes and define mutations linked to clinical outcomes. We also hypothesized that if HPV represented a driver in SNSCC observed mutational profiles would be shared across HPV-associated SNSCC, HNSCC and cervical cancer (CESC).

Here, we show that HPV-associated SNSCC exhibits mutational patterns similar to those seen in HPV-associated cervical and head and neck squamous cell carcinomas, including an APOBEC mutational signature and hotspot PI3K/FGFR3 mutations. We identify recurrent mutations (KMT2C, UBXN11, AP3S1, MT-ND4, MT-ND5) and find that KMT2D and FGFR3 mutations correlate with decreased overall survival. Additionally, we establish an HPV-associated SNSCC cell line and demonstrate that combinatorial inhibition of YAP/TAZ and PI3K pathways synergistically suppresses tumor cell clonogenicity. Altogether, our findings reveal distinct mechanisms of tumorigenesis in HPV-associated versus HPV-independent SNSCC and suggest that targeting YAP/TAZ and vertically inhibiting PI3K may be beneficial for HPV-associated tumors, whereas MYC-targeted approaches and horizontal inhibition of RAS/PI3K may be effective for HPV-independent disease.

## Results
### Cohort demographics and mutational analysis

The cohort consisted of 56 patients identified with squamous cell carcinoma arising from the sinonasal cavity or nasolacrimal duct. Of these, 37 (66.1%) were positive on RNA in situ hybridization for E6/E7 mRNA in high-risk HPV types (HR-HPV ISH), consistent with active transcription of viral oncogenes in tumor cells, and were thus termed HPV-associated. P16 status for HPV-associated tumors was positive in 34 (91.9%) patients, negative in 3 (8.1%) patients. Of HPV-independent tumors, p16 was positive in 3 (15.8%) patients, negative in 16 (84.2%) patients. As expected, the majority of HPV-associated tumors arose in the nasal cavity while the majority of HPV-independent tumors arose in the maxillary sinus (Table 1)[5]. As previously reported, patients with HPV-associated disease were younger at presentation (60.8 years) than HPV-independent (66.3 years) (Table 1, P < 0.05)[7].

Next, we performed high-throughput DNA sequencing of all HPV-independent and HPV-associated samples including cohorts with or without matched normal DNA. For HPV-independent SNSCC with matched normal DNA the most frequently mutated COSMIC cancer driver genes included TP53, NOTCH1, and KRAS (3/7 patients, 43%) as well as CDKN2A, COL2A1, FAT4, FBXW7, and ROS1 (2/7 patients, 29%) (Figs. 1A, and Supplementary Fig. 2A). Frequently mutated genes were then evaluated in our cohort of 12 additional HPV-independent SNSCC for which matched normal DNA was not available. Similar mutational frequencies were seen for TP53, NOTCH1, CDKN2A, COL2A1, and FAT4 (Figs. 1B, and Supplementary Fig. 2B).

For HPV-associated SNSCC with matched normal DNA the most frequently mutated COSMIC cancer driver genes included KMT2D (4/12 patients, 33%), FGFR3 and KMT2C (3/12 patients, 25%), GOLGA5, TET1, and ARID1B (2/12 patients, 16.7%) (Figs. 2A, and Supplementary Fig. 3A). Frequently mutated genes were then evaluated in our cohort of 25 additional HPV-associated SNSCC for which matched normal DNA was not available. Similar mutational frequencies were seen for KMT2D, FGFR3, KMT2C, and ARID1B (Figs. 2B, and Supplementary Fig. 3B). Interestingly, recurrent in frame deletions were identified in UBXN11 in 6/37 patients (16.2%) as well as frequent recurrent mutations in MT-ND4, MT-ND5, and AP3S1 (Supplementary Figs. 3A, B). Collectively, these data demonstrate a significantly different mutational profile between HPV-independent and HPV-associated SNSCC. The tumor mutational burden (TMB) was not statistically different across groups and was relatively low with a median of 2.19 mutations per megabase (Mut/Mb) for HPV-independent and 3.19 Mut/Mb for HPV-associated SNSCC.

**Table 1 | Patient demographics**

| | HPV-independent SNSCC | HPV-associated SNSCC |
|---|---|---|
| **Age at diagnosis** | | |
| Mean, years | 66.3 | 60.8 |
| Range, years | 40–87 | 43–75 |
| **Sex** | | |
| Male | 13 (68.4%) | 24 (64.9%) |
| Female | 6 (31.6%) | 13 (35.1%) |
| **Race/ethnicity** | | |
| White | 14 (73.7%) | 28 (75.7%) |
| Black | 2 (10.5%) | 7 (18.9%) |
| Other | 3 (15.8%) | 2 (5.4%) |
| **Subsite** | | |
| Nasal cavity | 5 (26.3%) | 17 (45.9%) |
| Ethmoid sinus | 2 (10.5%) | 5 (13.5%) |
| Maxillary sinus | 8 (42.1%) | 6 (16.2%) |
| Nasolacrimal duct | 3 (15.8%) | 7 (18.9%) |
| Other | 1 (5.3%) | 2 (5.4%) |
| **P16 status** | | |
| Positive | 3 (15.8%) | 34 (91.9%) |
| Negative | 16 (84.2%) | 3 (8.1%) |
| **Smoking status** | | |
| Current | 3 (15.8%) | 5 (13.5%) |
| Former | 8 (42.1%) | 18 (48.6%) |
| Never | 8 (42.1%) | 14 (37.8%) |

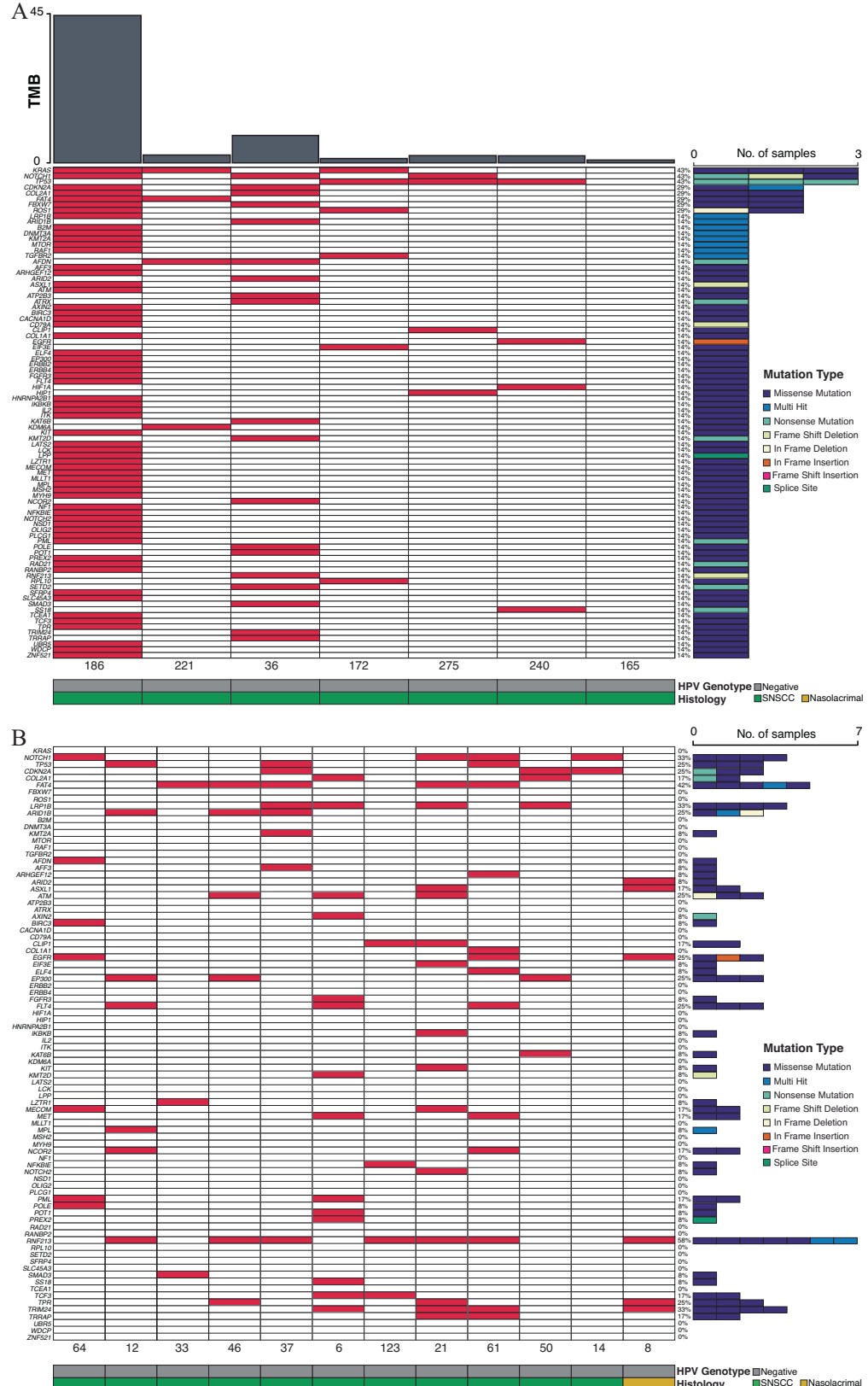

**Fig. 1 | High-throughput sequencing of HPV-independent SNSCC reveals mutational patterns in COSMIC genes. A** Whole-exome sequencing was performed in HPV-independent SNSCC with matched normal DNA (*n* = 7) and somatic variants were assessed. Mutations in COSMIC genes are represented. **B** Whole-exome sequencing was performed in HPV-independent SNSCC without matched normal DNA (*n* = 12) and somatic variants were assessed using a panel of normal genomes. Mutations in COSMIC genes are represented. Source data are provided as a **Source data** file.

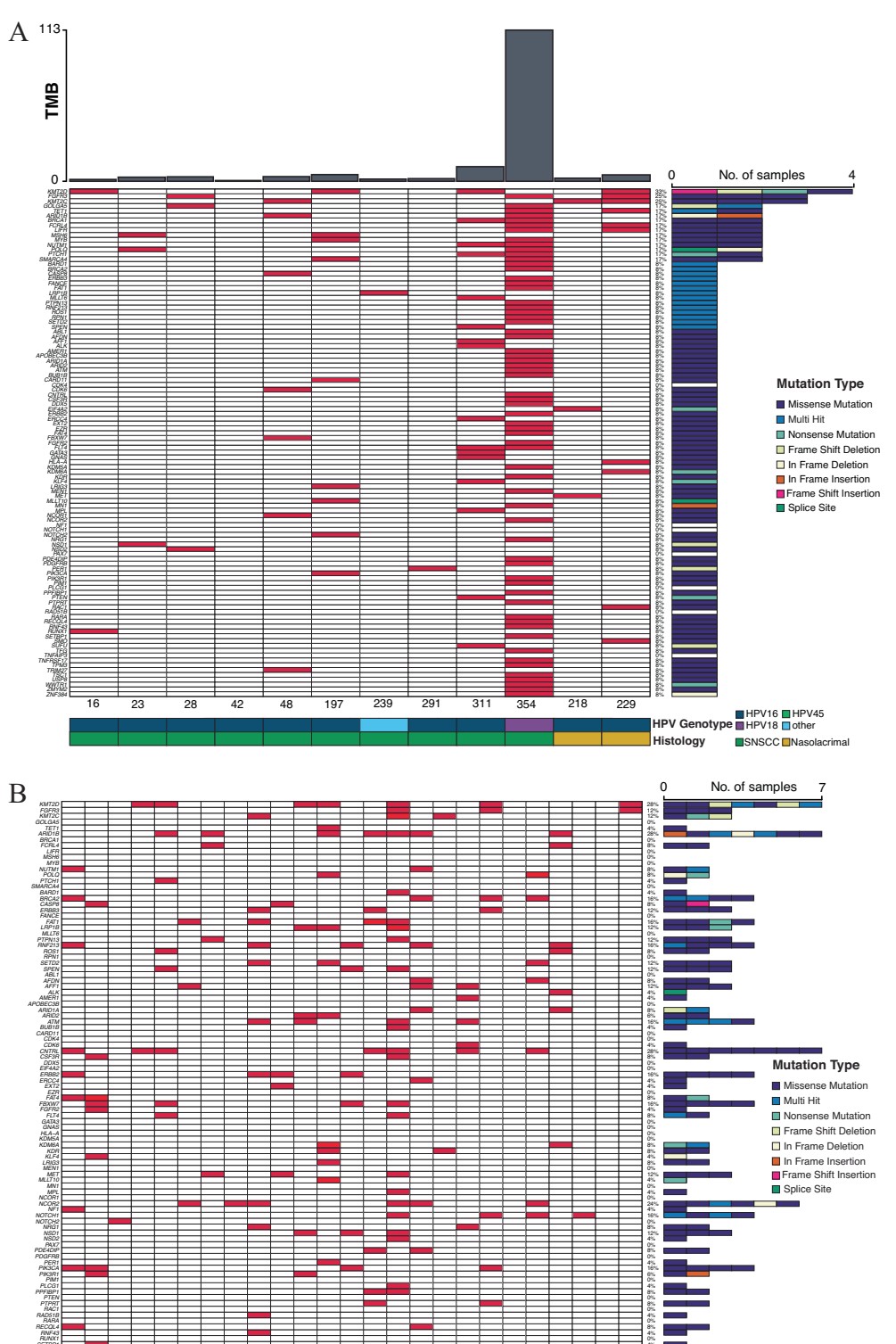

**Fig. 2 | High-throughput sequencing of HPV-associated SNSCC reveals mutational patterns in COSMIC genes. A** Whole-genome sequencing was performed in HPV-associated SNSCC with matched normal DNA ($n = 12$) and somatic variants were assessed. Mutations in COSMIC genes are represented. **B** Whole-genome sequencing was performed in HPV-associated SNSCC without matched normal DNA ($n = 25$) and somatic variants assessed using a panel of normal genomes. Mutations in COSMIC genes are represented. Source data are provided as a **Source data** file.

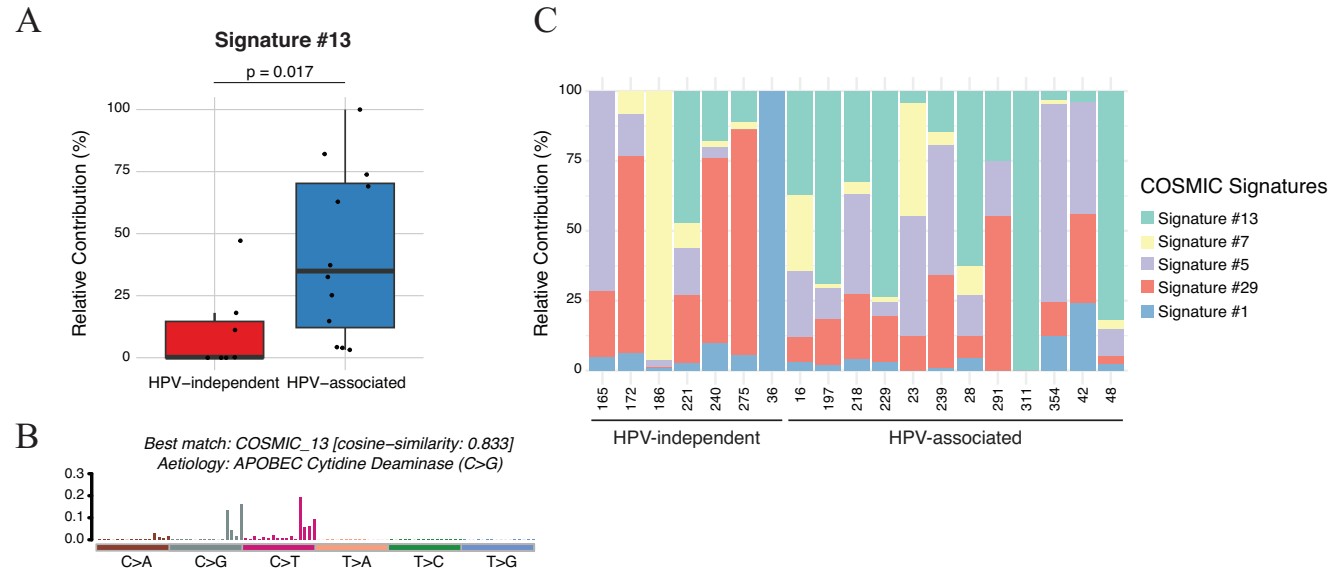

**Fig. 3 | APOBEC pathway signature 13 is enriched in HPV-associated SNSCC.**
**A** APOBEC signature 13 in the HPV-independent SNSCC compared to HPV-associated SNSCC cohort with matched normal DNA. Statistical significance was determined using an unpaired two-tailed *t*-test and a Mann-Whitney U test (*P* = 0.0171). The center line of the boxplot represents the median, the box spans the interquartile range (IQR, 25th to 75th percentile), and the whiskers extend to the minimum and maximum values within 1.5 × IQR. Outliers beyond this range are shown as individual points. Median values: HPV-independent = 0.001932 (*n* = 7), HPV-associated = 0.3492 (*n* = 12). **B** APOBEC signature 13 in the HPV-associated SNSCC cohort. **C** Stacked bar chart showing the relative contribution (%) of mutational signatures in individual HPV-associated and HPV-independent SNSCC samples. No significant differences were observed for signatures #1, 5, 7 and 29. Source data are provided as a **Source data** file.

## Identification of HPV-associated hotspot mutations

HPV-associated HNSCC and CESC have been characterized by hotspot mutations at E542K/E545K in *PIK3CA* and S249C in *FGFR3*[16–18]. Of the 5 HPV-associated SNSCC with *PIK3CA* missense mutations 3 tumors had E542K and 1 tumor had E545K mutations (4/5, 80%), while no HPV-independent SNSCC demonstrated *PIK3CA* hotspot mutations (Supplementary Data 1-2). Of the 6 HPV-associated SNSCC with *FGFR3* missense mutations 4 tumors had S249C hotspot mutations (4/6, 66.7%) while no HPV-independent SNSCC demonstrated *FGFR3* hotspot mutations. Interestingly, recurrent mutations were noted in HPV-associated SNSCC in *KMT2C* N729D (3/6, 50%), and *AP3S1* P158L (5/5, 100%) with recurrent in-frame deletions in *UBXN11* c.1464_1541 (6/8, 75%), and recurrent frameshift insertion/deletions or nonsense insertions in *MT-ND4* at c.237-238 and 4 additional individual sites (7/15, 46.7%), and in *MT-ND5* at c.891-892, c.1278-1279, and 4 additional individual sites (8/17, 47.1%) (Figs. 2A, B; and Supplementary Data 3–4). None of these mutations were observed in HPV-independent SNSCC (Fig. 1A-B; and Supplementary Data 1–2). Next, we evaluated the presence of SNSCC hotspot mutations in HNSCC and CESC in TCGA. Interestingly, while many *KMT2C* and *MT-ND5* mutations were noted in HNSCC or CESC, no mutations were noted at *KMT2C* N729D or *MT-ND5* c.891-892 or c.1278-1279 as seen in HPV-associated SNSCC (Supplementary Figs. 4A–D). Mutations in *AP3S1*, *UBXN11*, and *MT-ND4* were not frequently observed or were absent in HNSCC and CSCC, and none of the recurrent changes in these genes were seen as in HPV-associated SNSCC (Supplementary Figs. 5A–F).

Similar to HPV-associated HNSCC and CESC, mutation signature analysis revealed enrichment of APOBEC-associated mutational signature (COSMIC SB13) in HPV-associated SNSCC compared to HPV-independent SNSCC (Fig. 3A–C; *P* < 0.05). Even though a history of smoking was present in 11/19 (57.9%) of HPV-independent SNSCC and 23/37 (62.2%) of HPV-associated SNSCC, no enrichment in smoking mutational signature (COSMIC SBS4) was noted in either of these populations. Other signatures which did demonstrate enrichment did not demonstrate statistically significant differences when comparing

HPV-associated SNSCC to HPV-independent SNSCC (Supplementary Figs. 6A–B).

## Correlation of mutational status with overall survival

Next, we correlated clinical outcomes with the presence of frequent mutations in HPV-independent and HPV-associated SNSCC. As previously seen for HPV-independent HNSCC[19,20], the presence of a *TP53* mutation correlated significantly with worse overall survival in HPV-independent SNSCC (Fig. 4A, *P* < 0.01). No other statistically significant differences were noted with other genes of interest in HPV-independent SNSCC. Mutations in *KMT2D* and *FGFR3* in HPV-associated SNSCC were associated with worse overall survival (Fig. 4B, C, *P* < 0.05 and *P* = 0.0516, respectively) while a correlation was not seen for other genes of interest. To our knowledge, a correlation with *KMT2D* and survival has not been reported for HPV-associated HNSCC or CESC. We evaluated TCGA and found a correlation with *KMT2D* mutation and decreased overall survival in HPV-associated CESC (*P* < 0.05) but not HPV-associated HNSCC (*P* = 0.412) (Fig. 4D, E). For *FGFR3* mutations no significant difference in overall survival was noted for HPV-associated HNSCC and there were insufficient patients with *FGFR3* mutations in TCGA to evaluate for an association with CESC (Supplementary Fig. 7). Lastly, while HPV-associated SNSCC trended towards improved survival compared to HPV-independent SNSCC this did not reach statistical significance (*P* < 0.24; Supplementary Fig. 8).

## Assessment of viral integration

Another characteristic of HPV-associated HNSCC and CESC is frequent viral integration events in ~70–80% of HPV-associated tumors[21–23]. While many of these appear random or associated with common fragile sites, several hotspots have been identified near genes involved in epithelial stem cell maintenance including *FGFR*, *TP63*, *KLF5*, *SOX2*, and *MYC*[6,24]. We thus hypothesized that analysis of whole genome sequencing for HPV integration events may reveal similar patterns of viral integration in HPV-associated SNSCC as seen for HPV-associated

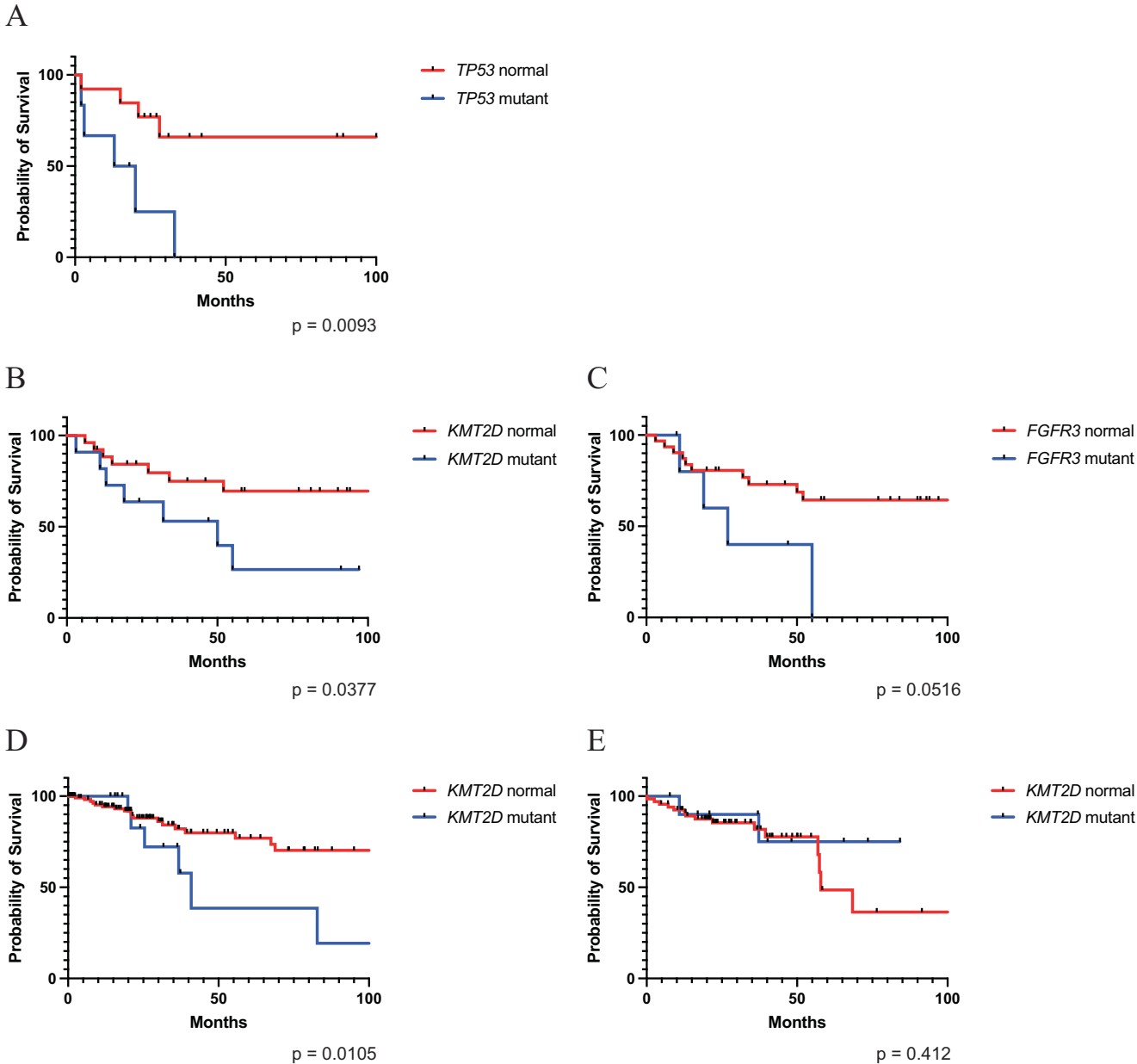

**Fig. 4 | Clinical associations with overall survival and mutation status. A** HPV-independent SNSCC with *TP53* mutations demonstrate an association with significantly decreased overall survival ($P = 0.0093$, Log-rank test: $\chi^2 = 6.757$, df = 1; *TP53* normal ($n = 13$), *TP53* mutant ($n = 6$)). **B** HPV-associated SNSCC with *KMT2D* mutations demonstrate an association with significantly decreased overall survival ($P = 0.0377$; Log-rank test: $\chi^2 = 4.316$, df = 1; *KMT2D* normal ($n = 26$), *KMT2D* mutant ($n = 11$)). **C** HPV-associated SNSCC with *FGFR3* mutations demonstrate an association with decreased overall survival ($P = 0.0516$; Log-rank test: $\chi^2 = 3.789$, df=1; *FGFR3* normal ($n = 31$), *FGFR3* mutant ($n = 6$)). **D** HPV-associated CESC from TCGA with *KMT2D* mutations demonstrate an association with decreased overall survival ($P = 0.0105$; Log-rank test: $\chi^2 = 6.54$, df = 1; *KMT2D* normal ($n = 123$), *KMT2D* mutant ($n = 17$)). **E** HPV-associated HNSCC from TCGA with *KMT2D* mutations and overall survival ($P = 0.4117$; Log-rank test: $\chi^2 = 0.6738$, df = 1; *KMT2D* normal ($n = 67$), *KMT2D* mutant ($n = 11$)). Survival curves were compared using a two-tailed Log-rank (Mantel–Cox) test. Source data are provided as a **Source data** file.

HNSCC and cervical cancer. Indeed, a wide variety of HPV integration events were identified across the genome (Figs. 5, and Supplementary Fig. 9, Supplementary Data 5). Upon closer evaluation we noted that two tumors had integration sites near *TP63* (patient #28) and *KLF5* (patient #60) (Supplementary Data 5). Out of the 27 observed integration events on autosomes, all overlapped fragile sites and integrations were observed in 2 samples in *FRA2S* (sample #2 and 26) and *FRA19B* (sample #11 and 311); and in 1 sample in *FRA1K* (sample #1) and *FRA6B* (sample #7). A surprisingly high enrichment for integration events were observed in SINE, LINE, and simple repeats; however, this was strongly skewed by a few samples including #11 and 218

(Supplementary Fig. 10). Lastly, we compared the distribution of integration events across the genome between SNSCC, HNSCC, and CESC demonstrating many similarities in integration event distribution (Supplementary Fig. 11).

**Elucidation of dysregulated pathways and molecular targets**

Next, we sought to integrate somatic coding mutation frequencies from HPV-associated and HPV-independent SNSCC with CNV analysis to elucidate dysregulated pathways and mechanisms of tumorigenesis. Significant PI3K pathway activation was observed in HPV-associated SNSCC, driven in large part by hotspot mutations in *PI3K* and *FGFR3* in

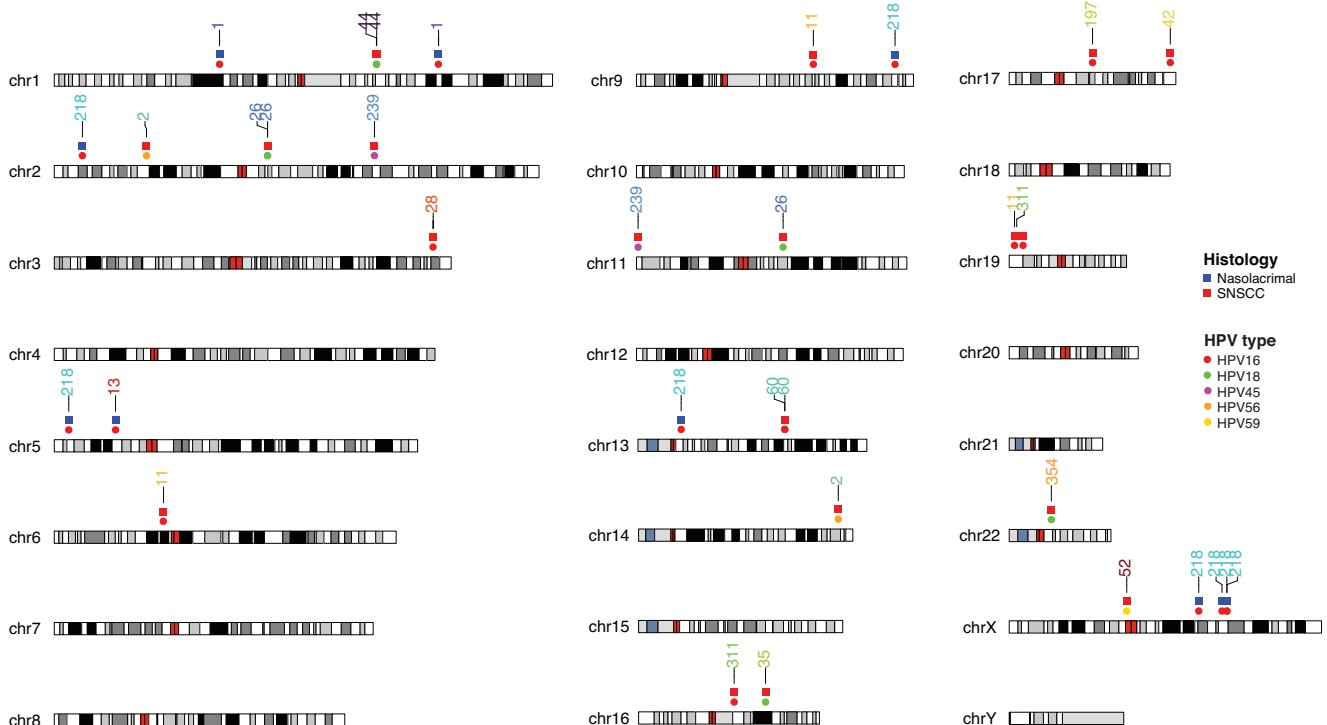

**Fig. 5 | Viral integration site analysis for HPV-associated SNSCC.** Viral integration sites for HPV-associated SNSCC are displayed for each chromosomal location. At each integration site the top row indicates the histology type while the bottom row indicates the HPV serotype. Source data are provided as a **Source data** file.

HPV-associated disease (Fig. 6). Interestingly, HPV-associated SNSCC also demonstrated activation of the YAP/TAZ pathway (driven predominantly by copy number amplification) as well as through inactivation of FAT family protocadherin inhibitors of the Hippo pathway (Fig. 6). The degree of YAP/TAZ pathway activation was comparable to HPV-associated HNSCC and HPV-associated CESC (Supplementary Fig. 12). When evaluating HPV-independent SNSCC, increased pathway activity was noted through both the PI3K and RAS pathways as well as MYC (driven predominantly by copy number amplification) (Fig. 6). Activation of RAS and MYC pathways was enhanced in HPV-independent SNSCC compared to HPV-independent HNSCC (Supplementary Fig. 13). Collectively, these results suggest that combinatorial blockade of YAP/TAZ and vertical inhibition of the PI3K pathway may be useful in targeting HPV-associated SNSCC whereas targeting MYC and horizontal inhibition of RAS/PI3K pathways for HPV-independent SNSCC[25].

We established a novel HPV-associated SNSCC cell line from patient #197, termed NCI-197 cells. This cell line was validated through both STR analysis as well as immunofluorescence staining for key markers of SNSCC (Supplementary Fig. 14). This patient's tumor and cell line demonstrated the *PIK3CA* E542K hotspot mutation. We assessed whether small molecule inhibition of PI3K using the small molecule inhibitor alpelisib may inhibit HPV-associated SNSCC cell proliferation. Indeed, NCI-197 cells were sensitive to alpelisib as assessed by both cell viability and cell impedance assays (Fig. 7A, B).

Based on mutational and CNV analyzes in HPV-associated SNSCC (Fig. 6) we hypothesized that combinatorial inhibition of PI3K and YAP/TAZ (verteporfin) may be particularly beneficial in targeting NCI-197 cells. YAP/TAZ are known to be important in stemness and tumor initiation, thus we assessed the clonogenicity of NCI-197 cells treated with small molecules inhibitors of the PI3K and TEAD, which disrupts a functionally essential protein interaction between YAP/TAZ-TEAD. Interestingly, while both alpelisib and verteporfin significantly reduced colony formation individually, combinatorial treatment at lower doses significantly reduced HPV-associated SNSCC colony formation in vitro

(Fig. 7C–E). As assessed by SynergyFinder the combinatorial effect was determined to be a strong synergistic interaction (Fig. 7F).

## Discussion

This study represents the most comprehensive genome-wide analysis of HPV-associated and HPV-independent SNSCC. Prior to these findings, it was unclear whether HPV functions as a tumorigenic factor in SNSCC or represents a bystander or contaminant in sinonasal malignancies[1,11]. We identify multiple key molecular features of HPV-driven HNSCC and CESC, including the absence of *TP53* mutations in p16 positive HPV-associated SNSCC, the presence of hotspot mutations in *PIK3CA* and *FGFR3*, APOBEC-associated mutational signature enrichment, HPV viral integration at known hotspot locations, and frequent mutations in epigenetic regulators[6,16,21,24]. These common features strongly suggest that HPV is driving HPV-associated SNSCC tumor biology rather than acting as a neutral bystander. Integration of the mutational and CNV analyzes revealed combinatorial inhibition of PI3K and YAP/TAZ may be effective in reducing HPV-associated SNSCC cell clonogenicity and colony formation which was supported by synergistic small molecule inhibition of these pathways with a newly-derived HPV-associated SNSCC cell line in vitro (Fig. 7).

HPV-associated SNSCC shares some features of both HPV-associated HNSCC and cervical cancer. As seen for HPV-associated HNSCC and CESC, hotspot mutations were frequently identified in HPV-associated SNSCC in *PIK3CA* and *FGFR3*. Interestingly, this study identified additional recurrent hotspot mutations in multiple additional genes in HPV-associated SNSCC. Recurrent in-frame deletions were observed in *UBXN11* (c.1464_1541) in HPV-associated SNSCC and may represent an additional potential mechanism of targeting *TP53* function. A UBXD family member, *UBXN2A*, has been found through a SEP domain to bind mortalin-2[26]. This binding results in unsequestration of p53, thus enhancing p53 activity. *UBXN11* is one of the few other UBXD family members which retains the SEP domain[26]. Therefore, mutations in *UBXN11* resulting in decreased activity or inappropriate cell localization could potentially lead to reduced p53 activity and be

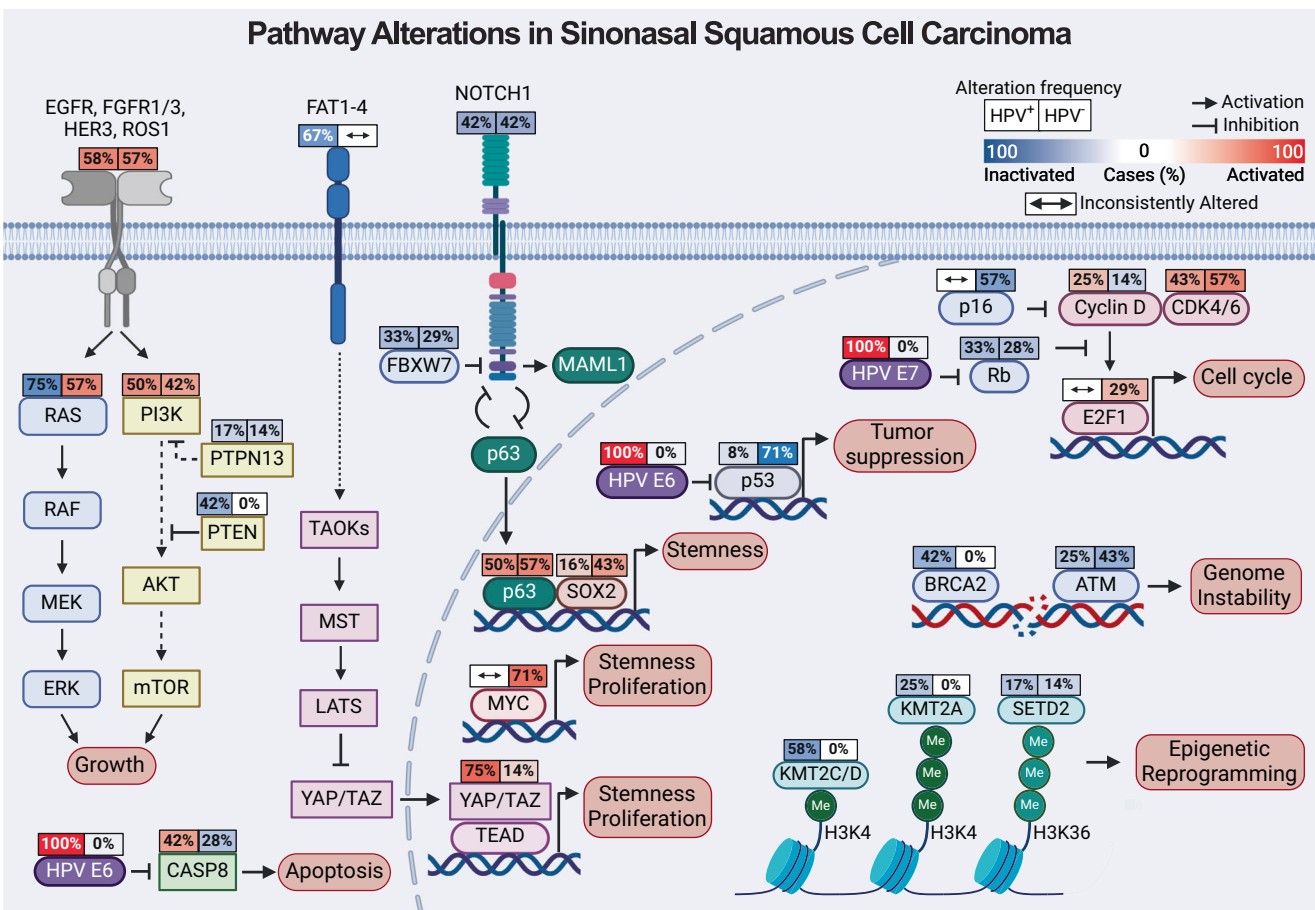

**Fig. 6 | Pathway alterations in sinonasal squamous cell carcinoma.** Incorporation of mutation and copy number alteration analysis reveals targetable pathways in HPV-associated SNSCC (left) and HPV-independent SNSCC (right). Source data are provided as a **Source data** file. Created in BioRender. Team, S. (2025) https://BioRender.com/r25r431.

an additional mechanism of p53 regulation in HPV-associated SNSCC; however, this hypothesis requires validation. Although the consequences of the *AP3S1* P158L recurrent mutation in SNSCC are not well understood, recent studies have linked *AP3S1* as a driver mutation in esophageal cancer as well as a potential pan-cancer oncogene through facilitation of an immunosuppressive tumor immune microenvironment[27,28]. *MT-ND5* and *MT-ND4* mutations have been noted to promote tumorigenesis and metastasis in other tumor types[29–31]. Collectively this study identifies multiple hotspot mutations which suggests these may be important oncogenic mechanisms in HPV-associated SNSCC due to their recurrent nature. However, this necessitates further investigation to determine their functional significance and biologic consequences.

The viral integration analysis in this study also identifies multiple key features of HPV-associated sinonasal malignancies including commonalities with HNSCC and CESC. Integration of HPV into or near *TP63* and *KLF5* is commonly observed across sinonasal malignancies, HNSCC, and CESC. Furthermore, all integration events on autosomes occurred at known common fragile sites. Additionally, significant enrichment of integration junctions was observed in SINE, LINE, and simple repeat sequences. Collectively these findings demonstrate many shared features of HPV integration as seen in HPV-associated HNSCC and CESC.

Viral integration analysis aided in ascertaining uncommon viral serotypes as the HPV serotype could be elucidated using this data for each HPV-associated tumor. As previously mentioned, there were 3 patients which were positive on HR-HPV ISH but negative for p16

(patient #30, 32, and 43) which were found to be positive for HPV types 114, 19, and 114, respectively. Interestingly, one of these three patients had a *TP53* missense mutation and another patient had a *TP53* nonsense mutation (Supplementary Data 4) and their overall survival was 9 and 13 months, respectively. These three patients also did not have any of the aforementioned recurrent mutations in *PIK3CA*, *FGFR3*, *KMT2C*, *AP3S1*, *UBXN11*, or *MT-ND5*. This suggests that SNSCC that are HR-HPV ISH positive but p16 negative have a tumor biology and behave more like HPV-independent SNSCC which may be an important clinical consideration. These data also further suggest that SNSCC should be tested for both HR-HPV ISH and p16.

There have been a few reports of HPV-associated nasolacrimal SNSCC; however, this group has not been well described due in part to their rare nature[32–34]. Our study identified 8 patients with SNSCC arising from the nasolacrimal system. Seven of eight (87.5%) were positive for HR-HPV ISH. These 7 tumors were p16 positive and HPV16 related. HPV-associated SNSCC of the nasolacrimal system had a similar demographic and mutational profile to HPV-associated SNSCC. These clinical and molecular features suggest that HPV-associated nasolacrimal SCC should be included in the HPV-associated SNSCC classification.

HPV-independent SNSCC has also been poorly studied outside of targeted *TP53* sequencing. The mutational signature of SNSCC-independent patterns demonstrates many parallels with HPV-negative HNSCC (Supplementary Fig. 13). *TP53* mutations were frequently observed but not to the same frequency as reported for HNSCC. Similarly, *TP53* mutated HPV-independent SNSCC had worse

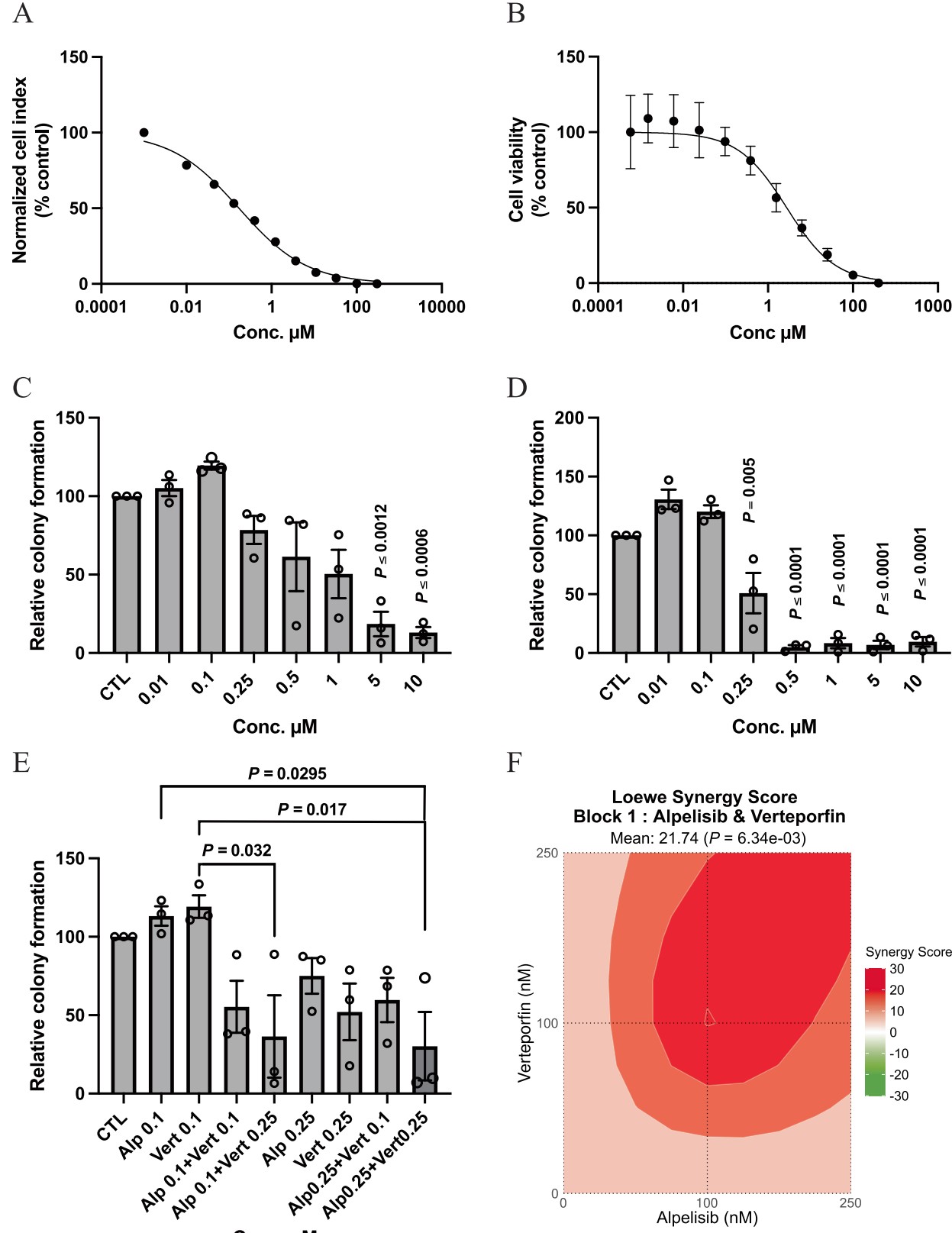

overall survival similar to prior reports for HPV-independent HNSCC (Fig. 4A). Furthermore, integration of mutational profiles and CNV data suggests that horizontal targeting of RAS/PI3K pathways and MYC may be an effective strategy for this tumor type (Supplementary Fig. 13).

While this study provides a comprehensive genomic analysis of HPV-associated and HPV-independent SNSCC, it is limited by relatively small numbers due to the rare nature of these tumors. The findings described in this study therefore require further investigation in future multi-institutional collaborative studies. Our decision to include eight nasolacrimal duct tumors was based on the direct anatomic connection and relationship of the nasolacrimal duct with the sinonasal cavity and prior evidence indicating a high prevalence of HPV16 in these

**Fig. 7 | Combinatorial blockade of PI3K and YAP/TAZ synergistically inhibits HPV-associated SNSCC clonogenicity in vitro. A** Cell viability analysis of NCI-197 cells 72 h after treatment with alpelisib by impedance assay. Data are (average, $n = 3$) representative of three independent experiments done in triplicates. **B** Cell viability analysis of NCI-197 cells 72 h after alpelisib treatment by ATP measurement. Data are mean ±SEM of three independent experiments ($n = 3$) done in triplicates. **C** Dose response of alpelisib and (**D**) Dose response of verteporfin assessed by colony formation assay of NCI-197 cells. Data represents mean ± SEM of three independent experiments ($n = 3$) done in triplicates. $P$-values were calculated by

ordinary one-way ANOVA with Tukey's multiple comparison test. Significant $p$-values are in comparison with control group. **E** Combination effects of alpelisib and verteporfin assessed by colony formation assay of NCI-197 cells. Data represents mean ± SEM ($n = 3$ per group) of three independent experiments done in triplicates. $P$-values were calculated by ordinary one-way ANOVA with Tukey's multiple comparison test. **F** Loewe synergy scores and $p$-values (bootstrapping of a dose-response matrix) calculated using SyngeryFinder software for alpelisib and verteporfin dose responses (100, 250 nM). Source data are provided as a **Source data** file.

lesions. By including nasolacrimal duct cases, we aimed to provide a more comprehensive understanding of HPV-associated malignancies in closely related anatomic subsites within the sinonasal region. Indeed, we found that HPV-associated nasolacrimal duct tumors shared key molecular features identified in this study with HPV-associated SNSCC. Secondly, due to the rare nature of these tumors the majority of our samples were retrospective and did not have matched normal DNA available requiring comparison to a panel of normal samples. Future studies will ideally include an increased number of samples with matched normal DNA and prospectively collected clinical information. Lastly, integration detection may also be influenced by the tools or criteria utilized.

## Methods

### Patient samples

This study complies with all relevant ethical regulations and was approved by the Institutional Review Boards at Johns Hopkins and the National Institutes of Health. Forty-two patients were retrospectively identified at Johns Hopkins from 2008 to 2020 with sufficient formalin-fixed paraffin-embedded (FFPE) tissue and a diagnosis of squamous cell carcinoma arising from the sinonasal cavity or nasolacrimal duct. In prior investigational studies we observed that the vast majority of squamous cell carcinoma arising from the nasolacrimal duct were positive for HPV16 and shared similar histopathological features to HPV-associated SNSCC. Given these observations and the close anatomical proximity of the nasolacrimal system to the nasal cavity and paranasal sinuses inclusion of nasolacrimal samples was planned a priori to provide a comprehensive analysis of SNSCC in the sinonasal region. Of these patients, 6 had neck dissection samples from which matched normal control DNA was obtained. Fourteen patients from 2020 to 2022 were prospectively enrolled, and written informed consent was obtained. Snap frozen tumor tissue was obtained along with peripheral blood mononuclear cells (PBMCs) to provide matched normal control DNA. Insufficient PBMCs for matched normal DNA comparison occurred for patient #123. All samples underwent HPV RNA in situ hybridization with a cocktail of high-risk HPV serotypes as previously described[7]. This was performed using a cocktail probe recognizing HPV E6/E7 mRNA from 18 high-risk HPV serotypes via RNAscope RNA in situ hybridization (Advanced Cell Diagnostics, Hayward, CA). Staining was evaluated by a board-certified pathologist and cases that demonstrated multiple nuclear and/or cytoplasmic signals visible at least 200x magnification were deemed positive. P16 staining was performed, and positivity was determined as previously described[7]. Positive staining was defined as strong, diffuse nuclear and cytoplasmic staining in ≥ 70% of tumor cells by a board-certified pathologist. Patients with inverted papilloma-related sinonasal squamous cell carcinoma were excluded from the study.

### DNA extraction

DNA was extracted from 20–30 mg frozen tissue samples using the AllPrep DNA/RNA Mini Kit (QIAGEN, 80204). Tissue was homogenized in 600 μL Buffer RLT Plus with β-mercaptoethanol (β-ME) using the TissueRuptor II (QIAGEN, 9002755). After centrifugation, the supernatant was transferred to AllPrep DNA spin columns. The column was

centrifuged at ≥ $8000 \times g$ for 30 sec. Genomic DNA was purified by adding 500 μL Buffer AW1, centrifuging, discarding the flow-through, applying 500 μL Buffer AW2, and centrifuging again. Finally, 100 μL Buffer EB was added, incubated for 5 min, and centrifuged for DNA elution. The DNA was stored at −20 °C.

The AllPrep DNA/RNA FFPE Kit (QIAGEN, 80234) was utilized to extract DNA from the FFPE (Formalin-Fixed Paraffin-Embedded), each containing 2-3 tissue cores. First, the samples were ground in 1.5 mL Eppendorf tubes using a disposable pestle (K7495211590). Then, de-paraffinization was performed by adding xylene, vortexing, and centrifuging to remove the supernatant without disturbing the pellet. Ethanol was used to remove residual xylene before incubating the sample until ethanol evaporation. The pellet was then resuspended in 150 μl Buffer PKD and 10 μl proteinase K, followed by incubation at 56 °C for 15 min and then on ice for 3 min. After centrifugation for 15 min at $20,000 \times g$, the supernatant was carefully transferred to a new tube, while the pellet was retained for DNA purification. For DNA extraction, the pellet was resuspended in 180 μl Buffer ATL and 40 μl proteinase K, vortexed, and incubated at 56 °C for 1 h, followed by incubation at 90 °C for 2 h without agitation. After a brief centrifugation, 200 μl Buffer AL and 200 μl ethanol (96–100%) were added to the sample, thoroughly vortexed, and transferred to a QIAamp MinElute spin column. The sample was then centrifuged for 1 min at ≥ $8000 \times g$. Subsequent washing steps with Buffer AW1, Buffer AW2, and ethanol were performed with centrifugations at ≥ $8000 \times g$ for 15 sec each. The spin column was centrifuged at full speed for 5 min to dry the membrane. DNA elution was achieved by adding 40 μl Buffer ATE onto the spin column membrane, incubating for 5 min at room temperature, and centrifuging at full speed for 1 min. The extracted DNA was stored at −20 °C.

PBMCs were prepared for DNA extraction using the QIAamp DNA Mini and Blood Mini (QIAGEN, 51104). The PBMC sample was mixed with QIAGEN Protease and Buffer AL and then incubated at 56 °C to release DNA. Ethanol was added to the lysate to promote DNA binding. The mixture was applied to QIAamp Mini spin columns and centrifuged to bind DNA to the membrane. The columns were washed with Buffer AW1 and AW2 to purify the DNA. Finally, DNA was eluted with Buffer AE and stored at −30 to −15 °C.

### Whole exome sequencing and whole genome sequencing

HPV-associated samples underwent whole genome sequencing (WGS, $n = 37$), while HPV-independent samples underwent whole exome sequencing (WES, $n = 19$). WGS was chosen for HPV-associated SNSCC to enable comprehensive detection of HPV integration events across the entire genome and characterization of non-coding regions that may be involved in viral integration and genome instability. WES was used for HPV-independent SNSCC to focus on coding mutations, given resource constraints and the primary interest in identifying somatic mutations in exonic regions. This approach allowed us to maximize the utility of our sequencing data for each cohort. Libraries for WES were constructed using the Agilent SureSelect V5 post-capture, beginning with a minimum of 200 ng of genomic DNA and the SureSelect[XT] Target Enrichment System for Illumina paired-end multiplexed sequencing. Libraries for WGS were constructed using TruSeq Nano

DNA Library Prep kit that requires minimum 100 ng of genomic DNA per sample. Paired-end sequencing was performed using the NovaSeq 6000 S4 Flow Cell with 2 × 150 cycle chemistry with an output of ~8 Gb (150X) per sample for WES or ~110 Gb (30X) per sample for WGS (Supplementary Data 6).

## Bioinformatics Analyzes

Sequencing data were demultiplexed using Illumina bcl2fastq2 (v.2.17.1.14) with default filters. Trimgalore (v0.6.7) was used to trim off adapter sequences and low-quality bases. Reads were then aligned against the hg38 genome using BWA-MEM (v0.7.17, Sentieon 202010.02 release). Duplicate reads were removed using Picard tools (v2.9.0, Sentieon 202010.02 release). Final recalibrated alignment files were created using Genome Analysis Toolkit (GATK, v3.8.0, Sentieon 202010.02 release). To determine coverage at different levels of partitioning and aggregation, Samtools depth v1.10 & GATK DepthOfCoverage v3.8.0 were used for WES data, while Bedtools genomecov v2.30.0 was used for WGS data. Somatic variants between the tumor-normal pairs were called using GATK MuTect2 (v3.8.0, Sentieon 202010.02 release). For samples without matched normal, the panel of normal genomes from the 1000 Genomes database provided by GATK was used. GATK HaplotypeCaller (v3.8.0, Sentieon 202010.02 release) was used to call germline variants in each sample. Passed somatic and germline variants were converted into Mutation Annotation Format files using vcf2maf (v1.6.19), and then summarized and visualized using maftools (v2.10.05). Copy number analyzes were performed using CNVkit (v0.9.4) for samples with matched normal, or Control-FREEC (v11.6) for samples without matched normal. Mutational signature analysis was performed using maftools (v2.18.0). The relative signature contribution barplot and boxplots were created using ggplot (v2_3.5.0). The mean comparison p-values were added to the boxplots using ggpubr (v0.6.0.999).

For mutation frequency analyzes, The Cancer Genome Atlas (TCGA) HNSCC[35] and CESC[22] datasets were extracted using cBioportal[36,37]. HNSCC tumor HPV status was based on RNAseq reads aligning to the HPV genome, yielding 449 HPV-negative and 78 HPV-positive HNSCC tumors of all anatomic sites[38,39]. For CESC tumors, only cervical squamous cell carcinoma (n = 141) was included. Cervical adenocarcinoma and cervical adenosquamous carcinoma as well as HPV-negative cervical tumors were excluded[22]. Cancer-associated signaling pathways were prioritized based on pathway constituent protein products of genes frequently altered in SNSCC, HNSCC, or CESC[22,35,40,41]. Genomic alteration frequency was defined as the proportion of samples with either somatic coding sequence alteration (single nucleotide alteration or short insertion/deletion) or copy number alteration ($\log_2$ copy number gain/loss > 0.3). Pathway figures were drawn using BioRender (biorender.com).

## HPV integration analysis

HPV integration analysis was conducted by first extracting the reads not mapping to the human genome or those with one unmatched mate-pair read. These reads were adapter and quality trimmed using fastp (0.23.2) remapped to a reference genome containing hg38 and all HPV genomes in the Papillomavirus Episteme (PaVE) as of 2018. From these alignments, the methods used in oncovirus_tools (https://github.com/gstarrett/oncovirus_tools, https://doi.org/10.5281/zenodo.3661416) were modified to determine integration sites for all detected HPV genomes. Integration sites were annotated using bedtools and the coordinates for all hg38 genes, fragile sites (https://webs.iiitd.edu.in/raghava/humcfs/), and repeatMasker elements. HPV types were called based on alignment and assembly data. For a type to be called, tumor sequencing must have had at least 10% of the HPV type genome covered with 2x or higher read depth. Concordance was checked against the de novo assemblies, which were matched to HPV types using blast taking the lowest e-value match (maximum cutoff of

1e-10) and the HPV type. Integration site enrichment near repeat elements and fragile sites was calculated by generating 1000 random integration events on the mappable hg38 genome. Chi-square calculation of observed versus expected integrations near repeat elements and fragile sites and were Bonferroni corrected for multiple comparisons. For integration site distribution and HPV type comparison across SNSCC, HNSCC, and CESC, data were retrieved[35] and from (http://www.vis-atlas.tech/) and cell lines and precancerous lesions were excluded.

## Establishment of HPV-associated SNSCC cell line

HPV-associated SNSCC tumor tissue was collected in PBS during surgical resection (patient #197, female). Tumor tissue was cleaned and cut into small fragments with a fine scissors and a single cell suspension using human tumor dissociation kit was performed per manufacturer instructions (Miltenyi Biotech., 130-095-929). After tumor dissociation, red blood cells (RBCs) were lysed with ACK lysing buffer and the cells were passed through 70 µM strainer. The resulting cell pellet was re-suspended in 70% ice cold basement membrane extract (Cultrex reduced growth factor BME type 2, Bio-Techne, 3533-010-02) in organoid growth media. Droplets of 50 µl were dispensed on the bottom of preheated 24 well plates. After incubating the plate at 37 °C for about 20–30 mins, 500 µl of pre-warmed patient-derived organoid media was added and returned to the incubator. Patient-derived organoid media consisted of 50% L-WRN conditioned media (obtained from ATCC and prepared as per instructions, CRL-3276) 50% advanced DMEM/F12 supplemented with 1X-B27 (life technologies, 17504-044), 1X penicillin-streptomycin (Thermo Fisher Scientific), 1.25mM N-acetyl-L-Cysteine (Sigma-Aldrich), 10 mM nicotinamide (Sigma-Aldrich), 0.5 µM A83-01 (Selleck Chemicals), 1 µM SB202190, 1 µM prostaglandin E2 (Selleck Chemicals), 1 µM Forskolin (Selleck Chemicals), 0.3 µM CHIR99021 (Sigma-Aldrich), 50 ng/ml hEGF, 10 ng/ml FGF10 and 5 ng/ml FGF2 (PeproTech). After culturing organoids in BME for several passages, the organoids attached to the bottom of the plate were subsequently cultured and expanded as a 2D cell line. The resulting patient-derived cell line was characterized by immunofluorescence staining of P63 (Santa Cruz Biotechnology, Sc-25268), P40 and cytokeratin AE1/AE3 (Santa Cruz Biotechnology, Sc-81714) and STR analysis was performed at Johns Hopkins Genetic Resources Core Facility. The generated cell line revives after cryopreservation and was only used for experiments up to 20 passages.

For cell viability assays, 10,000 cells were seeded in 96 well plates and different doses of alpelisib were added following overnight incubation. After 72 h, cell viability was assessed using CellTiter-Glo 2.0 (Promega, G9241) per manufacturer instructions. Data was normalized to (100%) vehicle control and baseline (0%) 0.1% Triton X-100 and kill curves produced by fitting lines with log inhibitor versus normalized response-variable slope using Graph-Pad Prism software. Realtime impedance assay was performed as previously described[42]. For colony formation clonogenicity assays, 5000 cells were seeded in 12 well plates, allowed to attach overnight and then cells were incubated with different doses of alpelisib (Selleck Chemicals) or verteporfin (Selleck Chemicals) separately and in combination for 72 h. After 72 h, media was changed, and cells were grown for a week and stained with crystal violet. Plates were scanned and the colonies were counted.

## Statistical analysis

For comparisons between two groups an unpaired, two-tailed, t-test was used. Log-rank (Mantel-Cox) tests were used to determine statistical significance for survival analyzes. A p-value significance threshold of $P < 0.05$. Synergy scores were calculated using SynergyFinder (Netphar, University of Helsinki, Helsinki, Finland) with a synergy score greater than 10 indicating a strong synergistic interaction[43]. Graphs were prepared using GraphPad Prism version 10.1.1.

**Reporting summary**

Further information on research design is available in the Nature Portfolio Reporting Summary linked to this article.

## Data availability

Sequencing data generated in this study for patients with informed consent allowing public sharing have been deposited in the dbGaP database under accession code phs003591.v1.p1. The genomic data are available under restricted access due to patient privacy regulations and institutional ethics policies. Access can be obtained by submitting a data access request through dbGaP, which will be reviewed by the relevant institutional review board. The raw sequencing data are protected and not publicly available due to data privacy laws. The remaining sequencing data for these samples can be accessed upon request from the corresponding author, subject to a completed data transfer agreement. Due to ethical approval restrictions, unrestricted access to the raw data for these samples is not permitted. Data from the first cohort has been uploaded in dbGaP, while data deposition is not allowed with the second cohort. Mutation frequency analyzes were performed using publicly available TCGA HNSCC [https://www.cbioportal.org/study/summary?id=hnsc_tcga] and TCGA CESC datasets, accessed via cBioPortal. Source data are provided with this paper. The remaining data are available within the Article, Supplementary Information or Source Data file. Source data are provided with this paper.

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

## Acknowledgements

The authors would like to acknowledge and thank the Johns Hopkins Experimental and Computational Genomics Core for sequencing and bioinformatics expertise. This research was supported (in part) by the Intramural Research Program of the NIH, Center for Cancer Research, National Cancer Institute. This study was supported in part by a research grant from Investigator-Initiated Studies Program of Merck Sharp & Dohme, LLC (N. London). The opinions expressed in this paper are those of the authors and do not necessarily represent those of Merck Sharp & Dohme, LLC. All other authors declare no competing interests. This study was presented at the European Skull Base Society Meeting in Maastricht, Netherlands on June 6, 2024.

## Author contributions

F.T.Z., S.G., G.J.S. and N.R.L. conceptualized the study and designed experiments; F.T.Z., S.G., and K.V. performed experiments; F.T.Z., G.J.S., F.F., T.T., A.G., Y.Z., D.L.F., M.E.B., T.G., and N.R.L. performed sequencing or bioinformatics analyzes; F.T.Z., A.S., N.R.R., M.R.J., A.P.L., C.F., G.L.G., L.M.R. and N.R.L. collected clinical samples or clinical data; F.T.Z., S.G., G.J.S., F.F., T.T., A.S., K.V., A.G., Y.Z., D.L.F., M.E.B., T.G., C.F., G.L.G., C.T.A., L.M.R., and N.R.L. interpretation of data; F.T.Z., S.G., G.J.S. and N.R.L. wrote the initial manuscript draft. All authors contributed to critical review and approval of the manuscript.

## Funding

## Competing interests

This research was supported (in part) by the Intramural Research Program of the NIH, Center for Cancer Research, National Cancer Institute. This study was supported in part by a research grant from Investigator-Initiated Studies Program of Merck Sharp & Dohme, LLC (N. London). The opinions expressed in this paper are those of the authors and do not necessarily represent those of Merck Sharp & Dohme, LLC. All other authors declare no competing interests.

## Additional information

[1]Department of Otolaryngology - Head and Neck Surgery, Johns Hopkins University School of Medicine, Baltimore, MD, USA. [2]Sinonasal and Skull Base Tumor Section, Surgical Oncology Program, Center for Cancer Research, National Cancer Institute, National Institutes of Health, Bethesda, MD, USA. [3]Center for Cancer Research, National Cancer Institute, National Institutes of Health, Bethesda, MD, USA. [4]Department of Otolaryngology - Head and Neck Surgery and Moores Cancer Center, University of California San Diego Health, La Jolla, CA, USA. [5]Sidney Kimmel Comprehensive Cancer Center, Johns Hopkins University, School of Medicine, Baltimore, MD, USA. [6]Department of Otolaryngology - Head and Neck Surgery, Harvard Medical School, Boston, MA, USA. [7]Department of Neurosurgery, Johns Hopkins University School of Medicine, Baltimore, MD, USA. [8]Department of Oncology, Johns Hopkins University School of Medicine, Baltimore, MD, USA. [9]Head and Neck Section, Surgical Oncology Program, Center for Cancer Research, National Cancer Institute, National Institutes of Health, Bethesda, MD, USA. [10]Department of Pathology, Johns Hopkins University School of Medicine, Baltimore, MD, USA. [11]These authors contributed equally: Fernando T. Zamuner, Sreenivasulu Gunti, Gabriel J. Starrett. ✉e-mail: nyall.london@nih.gov

