## [Transparent Peer Review file · Nature Communications]

Molecular patterns and mechanisms of tumorigenesis in HPV-associated and HPV-independent sinonasal squamous cell carcinoma

Corresponding Author: Professor Nyall R London Jr

Version 0:

Reviewer comments:

Reviewer #1

(Remarks to the Author)

Zamuner et al have performed a genome-wide profiling of 56 sinonasal squamous cell carcinoma (SNSCC) patients through high-throughput sequencing. The authors did WES for HPV-independent SNSCC and WGS for HPV-associated samples. They evaluated patterns of somatic mutations in genes, survival with key mutations, viral integration distribution, and dysregulated pathways between HPV-independent and -associated samples. In the end, they developed an HPV-associated cell line through the inhibition of PI3K and YAP/TAZ. Since there are no similar studies on such a large SNSCC cohort, I think the molecular patterns are necessary and important. However, I still have some questions that hopefully could help improve the manuscript.

Major comments:

1. the authors aim to show HPV drives SNSCC tumorigenesis. They found distinct mutations and enriched pathways between HPV-associated and HPV-independent SNSCC samples. They also show some similarities with other HPV-associated carcinoma. However, there is still a lack of direct evidence for the causality of HPV infection in SNSCC. For instance, the authors should show the expression of E6/7 in these samples and more functional analysis rather than characteristics, which is also associated with Figure 6.

2. it was unclear the HPV integration distribution and patterns between SNSCC and HNSCC and cervical cancer. The authors could perform a direct comparison. In addition, the integration detection could also be influenced by the different tools, and criteria used. It would also be interesting to know the cellular proportion or integration allele frequency of different integration sites. The number of supporting reads should also be given in the supplementary table.

Minor comments:

1. Page 4, line 73: There was no explanation for comparing OPSCC. The authors should include a sentence explaining the relationships between these carcinomas including HNSCC, SNSCC, OPSCC, HMSC.

2. Page 5, line 100: how the authors explained that TP53 was lacking in your HPV-associated samples but was also the only mutation reported in the HPV-associated sample in the two studies^{17,18}.

3. Page 6, line 115: what "HR-HPV ISH" represent?

4. Page 10, line 212: I can't see the enrichment of HPV integration events in repeats. Randomization and permutation tests should be done to compute the enrichment.

5. Page 19, line 419: As I know, the definition of HPV positive was usually based on the presence of HPV DNA reads in case the viral gene expression level is low. The definition should also be consistent throughout the manuscript.

6. Page 20, line 432: it was unclear the specific approach and criteria to distinguish the many HPV subtypes.

7. what is the order for the Figure 1 and 2? A and B panels could be merged with an annotation of with/without normal. Or combined Figures 1 and 2 to show the different patterns between HPV-associated and HPV-independent samples.

8. I was unclear about the conclusion of Figure 3C.

9. what is survival between HPV-positive and HPV-independent samples? And the same comparative analysis for other carcinomas.

10. since the mutations in HPV-associated and -independent samples were detected using WGS and WES separately. The authors should give a summary of the sequencing data.

Reviewer #2

(Remarks to the Author)

Thank you for submitting your manuscript "Molecular patterns and mechanisms of tumorigenesis in HPV-associated and HPV-independent sinonasal squamous cell carcinoma" to our journal. I enjoyed reading your work and congratulate you and your team in developing a HPV-associated SNSCC cell line. I have the following comments:

1) Methods: the reference number 7 (line 353 and 354) is a SEER database study and does not describe the RNA ISH and p16 staining that were performed for this study. I assume the authors used RNAscope for HR-HPV mRNA E6/E7 ISH and some form of IHC staining for p16. These are qualitative staining that warrants clarification (grading in staining intensity, percentage of stains deemed positive)

2) Methods: an explanation on why different methods of sequencing were chosen is warranted for the two cohorts (WGS for HPV-associated SNSCC, and WES for HPV-independent SNSCC)

3) Methods: It's not entirely clear to me why HMSC was chosen to be included in this study. The title of manuscript suggests it's an investigation of SNSCC, but HMSC is more closely associated with adenoid cystic carcinoma PMID: 28877065. I understand the relationship of HPV makes it enticing to include this group of pathology, but there are also other HPV-associated sinonasal carcinomas that are omitted (namely sinonasal undifferentiated carcinomas PMID: 24421248, PMID: 32416210). For the sake of clarity I would ask the authors to omit this cohort from this study and report it separately.

4) Results: If the authors are painting the narrative that there is a survival benefit in HPV-associated SNSCC, then a KM overall survival curve of the entire cohort, then stratified by HPV status should be included in Figure 4

5) Methods: 1 patient with HPV-associated SNSCC and 1 patient with HPV-independent SNSCC had "unknown" p16 status. Please explain.

6) Methods: This is a study with very limited samples, to count 56 patients one would have to include the 8 HMSC samples that should be omitted from analysis. To include 8 additional samples from the nasolacrimal system is questionable- the design of the study would naturally omit unusual anatomic sites outside of nasal cavity and paranasal sinuses. On the basis that their mutation signatures are consistent with the conventional cohort, I find it acceptable to include them in the analysis with the HPV-associated SNSCC cohort. Nevertheless, the authors should clarify in the methods whether this was planned or ad hoc.

7) Conclusion: The study's science is fundamentally sound if not limited in sample size or scope with the use of WGS/WES without RNAseq. However, I have reservations regarding "HPV is driving HPV-associated HPV-associated SNSCC tumor biology rather than acting as a neutral bystander." There's a bit of false equivalence: HPV-related SNSCC is not the same as HPV-related OPSCC, as evidenced by the heterogenous detection rate of HPV in the sinonasal tract dating back to 1980s. Various detection systems have been used including DNA ISH, GP5/GP6 PCR protocols that uses L1/L2 amplification, and most recently mRNA ISH that addresses the issue of viral integration without the evolutionary conserved segment. Nevertheless, the detection rate of HPV in sinonasal tumors varies tremendously, with 25% quoted to be overly optimistic, especially with very poor concordance rate with p16. This would seem to suggest that in tumors that were HPV positive by PCR (the most sensitive quantitative assay) that they may be episomal and may have a separate mechanism of pathogenesis PMID: 31743131 PMID: 28181187. Ultimately, I think the manuscript has made a compelling argument that there is a subset of SNSCC that has transcriptionally active HR HPV to have similar mutation signatures as HPV-associated OPSCC, but this number is a minority of SNSCC and the survival benefit is unclear PMID: 35454782. Interestingly, the authors opted to exclude SNSCC that were transformed from sinonasal inverted papilloma (SNIPs) that may have an association with HPV PMID: 29145573. This would have been a much more interesting cohort to include and whether they have a distinct mutation signature either due to the fact that they arise from a benign entity or that they are more likely to have HPV-driven oncogenesis.

Reviewer #3

(Remarks to the Author)

- What are the noteworthy results?

The authors show for the first time that HPV may be driving tumorigenesis in SNSCC as it is known in other human mucosal surfaces (oropharynx/head-and-neck, cervix and more). They identify mutational profiles in known cancer related genes and

genome wide signatures (APOBEC) distinct from their controls and viral integration in common hotspots. The authors show that combinatorial small molecule inhibition of YAP/TAZ and PI3K pathways inhibit clonogenicity in an HPV+ SNSCC cell line they developed and propose means to target these cancers (YAP/TAZ blockade w/vertical PI3K inhibition) differing from targets in HPV-independent SNSCCs (MYC and horizontal inhibition of RAS/PI3K pathways).

- Will the work be of significance to the field and related fields? How does it compare to the established literature? If the work is not original, please provide relevant references.

Yes, the work is of significance to the cancer and virology fields and it is original. Originality from identifying that HPV may also be a driver of tumorigenesis in tissue not previously established vulnerable to HPV infections which could lead to cancer development. In addition to proposing targeted intervention, the work raises several interesting medical and scientific questions of which HPV life cycle in sinonasal squamous cells and viral tropism are just two. SNSCC adds to the list of body sites where HPV is driving cancer development where the cervical cancer imposes the greatest global health burden. Prophylactic vaccines exists.

- Does the work support the conclusions and claims, or is additional evidence needed?

The study design is case-control with a further subdivision of the cases. The cases are High risk HPV dependent SNSCCs divided in HPV dependent SNSCCs and HPV related multiphenotypic sinonasal carcinoma (HMSC) and the controls HrHPV independent SNSCCs.

Due to the rarity of SNSCCs there are not so many samples in the cohort (56), yet the genomic analysis provides high resolution to the analysis. Yes, the work supports the conclusions and claims and I do not see an obvious need for additional evidence.

However, in samples with viral integration it would be very interesting for the reader to see the viral breakpoints, particularly in the HMSC with high HPV33 prevalence. Breakpoints in the viral E1/E2 genes are known drivers of tumorigenesis as negative regulation of viral oncogenes (E6 and E7) is thereby removed. One may hypothesise that the distinct features observed in the HMSC relative to the HPV-dependent SNSCCs could be explained by integration with such effective breakpoints.

- Are there any flaws in the data analysis, interpretation and conclusions? - Do these prohibit publication or require revision?

The study employs standard statistical and analytical methods for comparing groups, Log-rank for significance for survival and synergy (SynergyFinder) and seems appropriate.

- Is the methodology sound? Does the work meet the expected standards in your field?

Yes. I was curious if the use of matched samples vs. reference genomes in the WGS and WES could bias the data in any direction but do not find this and the methodologies are used in both cases and controls. The work meets the standards in the field.

- Is there enough detail provided in the methods for the work to be reproduced?

Yes.

Version 1:

Reviewer comments:

Reviewer #1

(Remarks to the Author)

The authors have addressed all of my concerns!

Reviewer #2

(Remarks to the Author)

I have reviewed the manuscript and the authors responses which were satisfactory.

Reviewer #3

(Remarks to the Author)

Thank you for including Supplementary Figure 9 with breakpoints.

RESPONSE TO REVIEWERS

We sincerely thank the reviewers for their thorough evaluation of our manuscript, “*Molecular patterns and mechanisms of tumorigenesis in HPV-associated and HPV-independent sinonasal squamous cell carcinoma.*” We appreciate their insightful comments and suggestions, which have significantly improved the quality and clarity of our work. Below, we address each of the reviewers' comments in detail.

REVIEWER COMMENTS

Reviewer #1 (Remarks to the Author):

Zamuner et al have performed a genome-wide profiling of 56 sinonasal squamous cell carcinoma (SNSCC) patients through high-throughput sequencing. The authors did WES for HPV-independent SNSCC and WGS for HPV-associated samples. They evaluated patterns of somatic mutations in genes, survival with key mutations, viral integration distribution, and dysregulated pathways between HPV-independent and -associated samples. In the end, they developed an HPV-associated cell line through the inhibition of PI3K and YAP/TAZ. Since there are no similar studies on such a large SNSCC cohort, I think the molecular patterns are necessary and important. However, I still have some questions that hopefully could help improve the manuscript.

Major comments:

1. The authors aim to show HPV drives SNSCC tumorigenesis. They found distinct mutations and enriched pathways between HPV-associated and HPV-independent SNSCC samples. They also show some similarities with other HPV-associated carcinoma. However, there is still a lack of direct evidence for the causality of HPV infection in SNSCC. For instance, the authors should show the expression of E6/7 in these samples and more functional analysis rather than characteristics, which is also associated with Figure 6.

Response: We thank the reviewer for this comment and the opportunity to improve our study. Every tumor sample in our study underwent high-risk HPV E6/E7 RNAscope RNA in situ hybridization. RNA in situ hybridization (rather than DNA in situ hybridization) is critical to demonstrate active expression of oncogenic HPV genes and thus demonstrate direct evidence of causality of HPV infection in SNSCC. This evidence, combined with the absence of *TP53* mutations in p16 positive HPV-associated SNSCC, the presence of hot spot mutations in *PIK3CA* and *FGFR3*, APOBEC enrichment, and HPV viral integration at known hotspot locations strongly demonstrate the downstream impact of HPV and that HPV is driving HPV-associated SNSCC tumor biology. We have added the following additional details to the manuscript Methods (**page 15, lines 334-338**) and Results (**page 6, lines 111-113**) to clarify HPV E6/E7 expression in HPV-associated SNSCC:

- Methods: “This was performed using a cocktail probe recognizing HPV E6/E7 mRNA from 18 high-risk HPV serotypes via RNAscope RNA in situ hybridization (Advanced Cell Diagnostics, Hayward, CA). Staining was evaluated by a board-certified pathologist

and cases that demonstrated multiple nuclear and/or cytoplasmic signals visible at least 200x magnification were deemed positive.”

- Results: “Of these 37 (66.1%) were positive on RNA in situ hybridization for E6/E7 mRNA in high-risk HPV types (HR-HPV ISH), consistent with active transcription of viral oncogenes in tumor cells, and were thus termed HPV-associated.”

Additionally, we have added a new Supplementary Figure (**Supplementary Figure 1**) demonstrating representative high-risk HPV E6/E7 RNA in situ hybridization staining in HPV-associated SNSCC to demonstrate direct evidence for the causality of HPV infection in SNSCC.

2. It was unclear the HPV integration distribution and patterns between SNCC and HNSCC and cervical cancer. The authors could perform a direct comparison. In addition, the integration detection could also be influenced by the different tools, and criteria used. It would also be interesting to know the cellular proportion or integration allele frequency of different integration sites. The number of supporting reads should also be given in the supplementary table.

Response: We thank you for these comments. As requested, we performed additional analysis and have added a new Supplementary Figure (**Supplementary Figure 11**) comparing the distribution of integration events between SNCC, HNSCC, and cervical cancer and added this point to the manuscript Results (**page 9, lines 202-204**) and Methods (**page 20, lines 437-439**). We have added to the manuscript limitations (**page 15, lines 315-316**) that “integration detection may also be influenced by the tools or criteria utilized”. Lastly, we have updated **Supplementary Table 5** detailing the integration sites and the supporting evidence along with relative virus copy information, as requested.

Minor comments:

1. Page 4, line 73: There was no explanation for comparing OPSCC. The authors should include a sentence explaining the relationships between these carcinomas including HNSCC, SNSCC, OPSCC, HMSC.

Response: We have added the following statement in the manuscript introduction to clarify the relationships:

- **Introduction (page 4, lines 73–77):** “Oropharyngeal squamous cell carcinoma (OPSCC) is a subset of HNSCC arising from the oropharynx and is well-known for its association with HPV. Similarly, SNSCC arises from the sinonasal tract, and understanding the parallels and distinctions among HNSCC subtypes including OPSCC and SNSCC is essential for elucidating the role of HPV in SNSCC tumorigenesis.”

Of note, we have removed reference to HMSC per Reviewer 2’s comments.

2. Page 5, line 100: how the authors explained that TP53 was lacking in your HPV-associated samples but was also the only mutation reported in the HPV-associated sample in the two studies^{17,18}.

Response: We included these studies in the manuscript introduction to report the only known report of WES of a single case of HPV-associated SNSCC and highlight the importance of conducting our study using a large cohort of SNSCC. There are several likely reasons for this observation:

First, in contrast to Stransky et al. (2011), which reports the results from this single case as part of a larger study, and Llorente et al. (2014), which is a review that mentions this single case, our study included a large cohort of HPV-associated SNSCC patients, providing a more comprehensive assessment of *TP53* mutation frequency in this group. In contrast, Stransky et al. included only a single case making it challenging to draw definitive conclusions about HPV-associated SNSCC mutational patterns.

Secondly, in our study we utilized high-risk HPV RNA in situ hybridization targeting E6/E7 **mRNA**, which confirms transcriptionally active HPV infection. This is in contrast to Stransky et al. which utilized **DNA** in situ hybridization which while very sensitive may detect artifacts of HPV infection or HPV which is not transcriptionally active, potentially leading to misclassification of HPV status.

Lastly, we did observe three patients in our cohort with high-risk HPV positive on RNA in situ hybridization (with uncommon high-risk HPV serotypes) but p16 negative (**Discussion, page 13, lines 283-287**). Of these three patients, two were found to have *TP53* mutations.

3. Page 6, line 115: what “HR-HPV ISH” represent?

Response: We have clarified the abbreviation upon first use in **Results (page 6, lines 111-112)** “RNA in situ hybridization for E6/E7 mRNA in high-risk HPV serotypes (HR-HPV ISH)”

4. Page 10, line 212: I can’t see the enrichment of HPV integration events in repeats. Randomization and permutation tests should be done to compute the enrichment.

Response: Thanks for this comment. We have conducted additional analysis and included a new Supplementary Figure demonstrating HPV integration enrichment (**Supplementary Figure 10**). Additional details have been added to the Methods (**page 20, lines 434-437**).

5. Page 19, line 419: As I know, the definition of HPV positive was usually based on the presence of HPV DNA reads in case the viral gene expression level is low. The definition should also be consistent throughout the manuscript.

Response: We appreciate the reviewer's attention to the definition of HPV positivity and the need for consistency. In our study, HPV positivity was defined based on the detection of E6/E7 mRNA using high-risk HPV RNA in situ hybridization (HR-HPV ISH). This method confirms transcriptionally active HPV infection, which is critical for establishing the role of the virus in oncogenesis. We acknowledge that some studies define HPV positivity based on the presence of HPV DNA reads, particularly when viral gene expression levels are low. However, detecting viral mRNA provides stronger evidence of active infection and oncogenic potential.

Regarding the TCGA datasets referenced in our manuscript, HPV status in HNSCC tumors was determined based on RNA sequencing (RNA-seq) reads aligning to the HPV genome, effectively detecting HPV transcripts (Refs. 41,42). This approach aligns with our method of defining HPV positivity based on viral mRNA expression. By aligning our definition with methods used in TCGA and ensuring consistent application throughout the manuscript, we maintain clarity and accuracy in our comparisons and analyses.

6. Page 20, line 432: it was unclear the specific approach and criteria to distinguish the many HPV subtypes.

Response: Thanks for this clarifying question. We have added the following statement to the Methods (lines 430-434):

“HPV types were called based on alignment and assembly data. For a type to be called, tumor sequencing must have had at least 10% of the HPV type genome covered with 2x or higher read depth. Concordance was checked against the de novo assemblies, which were matched to HPV types using blast taking the lowest e-value match (maximum cutoff of 1e-10) and the HPV type.”

7. what is the order for the Figure 1 and 2? A and B panels could be merged with an annotation of with/without normal. Or combined Figures 1 and 2 to show the different patterns between HPV-associated and HPV-independent samples.

Response: We appreciate the reviewer's suggestion to merge Figures 1 and 2 for a direct comparison between HPV-associated and HPV-independent SNSCC samples and we had considered this when writing the manuscript. However, we believe that keeping these figures separate enhances clarity and allows us to present the unique mutational landscapes of each cohort more effectively. In summary, 1) separate figures enable us to highlight the distinct mutational profiles and frequently mutated genes specific to each group without overcrowding the visual presentation; 2) HPV-independent SNSCC samples were analyzed using whole-exome sequencing (WES), while HPV-associated SNSCC samples underwent whole-genome sequencing (WGS). Combining data from different sequencing platforms might introduce confusion; 3) The cohorts differ in the number of samples and the proportion with matched normal DNA. Presenting them separately maintains consistency and provides appropriate context for observed mutation frequencies; 4) Merging all panels could result in a complex and crowded figure, making it difficult for readers to discern key findings and differences between the cohorts. We also revised the TMB calculation and updated the figures to express TMB in terms of mutations per megabase (mut/Mb). Specifically, these changes have been implemented in Figures 1, 2, and Supplementary Figures 2 and 3 to align with standard conventions and improve clarity.

8. I was unclear about the conclusion of Figure 3C.

Response: The purpose of Figure 3C is to provide further supportive representation of the data for individual samples that there is enrichment of the APOBEC mutational signature in HPV-associated SNSCC compared to HPV-independent SNSCC and no significant difference in other mutational signatures. We have amended the figure legend to improve clarity. This figure could

also be moved to (**Supplementary Figure 6**) along this other supportive representation of these data if deemed more appropriate by the Reviewers.

9. What is survival between HPV-positive and HPV-independent samples? And the same comparative analysis for other carcinomas.

Response: While our cohort did observe a trend towards improved survival in HPV-associated SNSCC this did not reach statistical significance ($P < 0.24$). We have included this new figure as **Supplementary Figure 8**. A previous large database study has demonstrated increased survival for HPV-associated SNSCC compared to HPV-independent SNSCC (Ref 10). The survival difference for other carcinomas has been thoroughly investigated by other groups including OPSCC (Ochoa et al., PMID: 36077856; **Figure 3**) and CESC (Cuschieri et al., PMID: 24740764; **Figure 1**).

10. Since the mutations in HPV-associated and -independent samples were detected using WGS and WES separately. The authors should give a summary of the sequencing data.

Response: Thanks for this question. We have added a new Supplementary Table (**Supplementary Table 6**) which provides a sample specific summary of the sequencing data and sequencing coverage.

Reviewer #2 (Remarks to the Author):

Thank you for submitting your manuscript "Molecular patterns and mechanisms of tumorigenesis in HPV-associated and HPV-independent sinonasal squamous cell carcinoma" to our journal. I enjoyed reading your work and congratulate you and your team in developing a HPV-associated SNSCC cell line. I have the following comments:

1) Methods: the reference number 7 (line 353 and 354) is a SEER database study and does not describe the RNA ISH and p16 staining that were performed for this study. I assume the authors used RNAscope for HR-HPV mRNA E6/E7 ISH and some form of IHC staining for p16. These are qualitative staining that warrants clarification (grading in staining intensity, percentage of stains deemed positive)

Response: We thank the reviewer for these comments. We recognize that the referenced methods in reference 7 may be challenging to find as they are located in the downloadable Supplement 1. To help clarify these important method details we have amended the methods section of our manuscript (**page 15, lines 334-340**) to read:

“This was performed using a cocktail probe recognizing HPV E6/E7 mRNA from 18 high-risk HPV serotypes via RNAscope RNA in situ hybridization (Advanced Cell Diagnostics, Hayward, CA). Staining was evaluated by a board-certified pathologist and cases that demonstrated multiple nuclear and/or cytoplasmic signals visible at least 200x magnification were deemed

positive. P16 staining was performed, and positivity determined as previously described⁷. Positive staining was defined as strong, diffuse nuclear and cytoplasmic staining in $\geq 70\%$ of tumor cells by a board-certified pathologist.”

2) Methods: an explanation on why different methods of sequencing were chosen is warranted for the two cohorts (WGS for HPV-associated SNSCC, and WES for HPV-independent SNSCC)

Response: We have added an explanation for our sequencing approach:

- **Methods (page 18, lines 378-384):** “WGS was chosen for HPV-associated SNSCC to enable comprehensive detection of HPV integration events across the entire genome and characterization of non-coding regions that may be involved in viral integration and genome instability. WES was used for HPV-independent SNSCC to focus on coding mutations, given resource constraints and the primary interest in identifying somatic mutations in exonic regions. This approach allowed us to maximize the utility of our sequencing data for each cohort.”

3) Methods: It's not entirely clear to me why HMSC was chosen to be included in this study. The title of manuscript suggests it's an investigation of SNSCC, but HMSC is more closely associated with adenoid cystic carcinoma PMID: 28877065. I understand the relationship of HPV makes it enticing to include this group of pathology, but there are also other HPV-associated sinonasal carcinomas that are omitted (namely sinonasal undifferentiated carcinomas PMID: 24421248, PMID: 32416210). For the sake of clarity I would ask the authors to omit this cohort from this study and report it separately.

Response: We understand the reviewer's concern regarding the inclusion of HMSC. As requested, we have removed HMSC from all manuscript and supplementary figures/tables (Manuscript Figures 2 and 5, Supplementary Figures 3 and 6, and Supplementary Tables 3 and 4) and will report the HMSC data separately in a future manuscript.

4) Results: If the authors are painting the narrative that there is a survival benefit in HPV-associated SNSCC, then a KM overall survival curve of the entire cohort, then stratified by HPV status should be included in Figure 4

Response: We thank the reviewer for this comment. The purpose of our study was not necessarily to paint a narrative of a survival benefit as this has been investigated in other larger database studies (Ref 10). While our cohort did observe a trend towards improved survival in HPV-associated SNSCC this did not reach statistical significance ($P < 0.24$). As this was not a major emphasis of the study and not a statistically significant difference, we elected to include this figure as a new **Supplementary Figure 8** rather than within Manuscript Figure 4.

5) Methods: 1 patient with HPV-associated SNSCC and 1 patient with HPV-independent SNSCC had "unknown" p16 status. Please explain.

Response: These two patient samples had not originally undergone p16 staining during clinical evaluation. In response to this question, we have now independently performed p16 staining for these two remaining samples. P16 was positive for the HPV-associated SNSCC sample and negative for the HPV-independent sample. These results have been updated in the manuscript and **Table 1**.

6) Methods: This is a study with very limited samples, to count 56 patients one would have to include the 8 HMSC samples that should be omitted from analysis. To include 8 additional samples from the nasolacrimal system is questionable- the design of the study would naturally omit unusual anatomic sites outside of nasal cavity and paranasal sinuses. On the basis that their mutation signatures are consistent with the conventional cohort, I find it acceptable to include them in the analysis with the HPV-associated SNSCC cohort. Nevertheless, the authors should clarify in the methods whether this was planned or ad hoc.

Response: The inclusion of nasolacrimal SNSCC samples was intentional and *a priori* based on investigational observations by our group and the anatomical proximity to the nasal cavity and paranasal sinuses. We have clarified this by including the following statement:

- **Methods (page 15, lines 322-328):** “In prior investigational studies we observed that the vast majority of squamous cell carcinoma arising from the nasolacrimal duct were positive for HPV16 and shared similar histopathological features to HPV-associated SNSCC. Given these observations and the close anatomical proximity of the nasolacrimal system to the nasal cavity and paranasal sinuses inclusion of nasolacrimal samples was planned *a priori* to provide a comprehensive analysis of SNSCC in the sinonasal region.”

7) Conclusion: The study's science is fundamentally sound if not limited in sample size or scope with the use of WGS/WES without RNAseq. However, I have reservations regarding "HPV is driving HPV-associated HPV-associated SNSCC tumor biology rather than acting as a neutral bystander." There's a bit of false equivalence: HPV-related SNSCC is not the same as HPV-related OPSCC, as evidenced by the heterogenous detection rate of HPV in the sinonasal tract dating back to 1980s. Various detection systems have been used including DNA ISH, GP5/GP6 PCR protocols that uses L1/L2 amplification, and most recently mRNA ISH that addresses the issue of viral integration without the evolutionary conserved segment. Nevertheless, the detection rate of HPV in sinonasal tumors varies tremendously, with 25% quoted to be overly optimistic, especially with very poor concordance rate with p16. This would seem to suggest that in tumors that were HPV positive by PCR (the most sensitive quantitative assay) that they may be episomal and may have a separate mechanism of pathogenesis PMID: 31743131 PMID: 28181187. Ultimately, I think the manuscript has made a compelling argument that there is a subset of SNSCC that has transcriptionally active HR HPV to have similar mutation signatures as HPV-associated OPSCC, but this number is a minority of SNSCC and the survival benefit is unclear PMID: 35454782. Interestingly, the authors opted to exclude SNSCC that were transformed from sinonasal inverted papilloma (SNIPs) that may have an association with HPV

PMID: 29145573. This would have been a much more interesting cohort to include and whether they have a distinct mutation signature either due to the fact that they arise from a benign entity or that they are more likely to have HPV-driven oncogenesis.

Response: We appreciate the reviewer's perspectives. We agree that while it is well accepted that HPV-positive tumor status confers a significant advantage to patients with OPSCC that this survival advantage is lower and less clear in HPV-associated SNSCC (PMID: 35454782 cited by the reviewer). The goal of our study, however, was not to specifically investigate whether HPV-associated SNSCC confers survival benefit compared to HPV-independent SNSCC, but to ascertain molecular mechanisms of tumorigenesis in HPV-associated and HPV-independent SNSCC. Interestingly, in regards to HPV-associated SNSCC we did find that mutations in *KMT2D* and *FGFR3* were associated with decreased survival (**Figure 4B-C**). Regarding the exclusion of SNSCC transformed from sinonasal inverted papilloma (IP-SNSCC), we agree that analyzing SNSCC arising from IP-SNSCC would be interesting, and we plan to investigate this in future studies. However, the majority of inverted papilloma arise from EGFR exon 20 insertion mutations whereas a minority of IP-SNSCC are associated with **low-risk** HPV 6/11 (PMID: 29145573). Low-risk HPV is governed by strikingly different mechanisms of tumorigenesis than high-risk HPV including substantially lower transforming activity of low-risk HPV E6/E7 and low-risk HPV E6 does not efficiently interact with p53 or induce p53 degradation (PMID: 15479788). Thus, we excluded IP-SNSCC from our present study which focused on mechanisms of SNSCC tumorigenesis in high-risk HPV.

Reviewer #3 (Remarks to the Author):

- What are the noteworthy results?

The authors show for the first time that HPV may be driving tumorigenesis in SNSCC as it is known in other human mucosal surfaces (oropharynx/head-and-neck, cervix and more). They identify mutational profiles in known cancer related genes and genome wide signatures (APOBEC) distinct from their controls and viral integration in common hotspots. The authors show that combinatorial small molecule inhibition of YAP/TAZ and PI3K pathways inhibit clonogenicity in an HPV+ SNSCC cell line they developed and propose means to target these cancers (YAP/TAZ blockade w/vertical PI3K inhibition) differing from targets in HPV-independent SNSCCs (MYC and horizontal inhibition of RAS/PI3K pathways).

- Will the work be of significance to the field and related fields? How does it compare to the established literature? If the work is not original, please provide relevant references.

Yes, the work is of significance to the cancer and virology fields and it is original. Originality from identifying that HPV may also be a driver of tumorigenesis in tissue not previously established vulnerable to HPV infections which could lead to cancer development. In addition to

proposing targeted intervention, the work raises several interesting medical and scientific questions of which HPV life cycle in sinonasal squamous cells and viral tropism are just two. SNSCC adds to the list of body sites where HPV is driving cancer development where the cervical cancer imposes the greatest global health burden. Prophylactic vaccines exist.

- Does the work support the conclusions and claims, or is additional evidence needed?

The study design is case-control with a further subdivision of the cases. The cases are High risk HPV dependent SNSCCs divided in HPV dependent SNSCCs and HPV related multiphenotypic sinonasal carcinoma (HMSC) and the controls HrHPV independent SNSCCs.

Due to the rarity of SNSCCs there are not so many samples in the cohort (56), yet the genomic analysis provides high resolution to the analysis. Yes, the work supports the conclusions and claims and I do not see an obvious need for additional evidence.

However, in samples with viral integration it would be very interesting for the reader to see the viral breakpoints, particularly in the HMSC with high HPV33 prevalence. Breakpoints in the viral E1/E2 genes are known drivers of tumorigenesis as negative regulation of viral oncogenes (E6 and E7) is thereby removed. One may hypothesise that the distinct features observed in the HMSC relative to the HPV-dependent SNSCCs could be explained by integration with such effective breakpoints.

Response: Thank you for your feedback and comments. We agree that analyzing viral integration breakpoints could provide valuable insights into the differences between HMSC and HPV-associated SNSCC. We have added a supplementary figure highlighting the detected breakpoints in the viral genomes and their overall coverage in HPV-associated SNSCC (**Supplementary Figure 9**). As per the request of Reviewer 2 the HMSC data have been removed from the present manuscript and will be separately reported in a future manuscript. At the request of Reviewer 3 we have generated the breakpoint analysis for HMSC (**Response to Reviewer Figure 1**) which will be included in a later manuscript in development.

[Redacted]

- Are there any flaws in the data analysis, interpretation and conclusions? - Do these prohibit publication or require revision?

The study employs standard statistical and analytical methods for comparing groups, Log-rank for significance for survival and synergy (SynergyFinder) and seems appropriate.

- Is the methodology sound? Does the work meet the expected standards in your field?

Yes. I was curious if the use of matched samples vs. reference genomes in the WGS and WES could bias the data in any direction but do not find this and the methodologies are used in both cases and controls. The work meets the standards in the field.

- Is there enough detail provided in the methods for the work to be reproduced?

Yes.

Response: We are thankful to the Reviewer for these comments.